Unexplained abdominal pain as a driver for inappropriate therapeutics: an audit on the use of intravenous proton pump inhibitors

Lai Pauline Siew Mei 1 2 plai@ummc.edu.my
Wong Yin Yen 2
Low Yong Chia 2
Lau Hui Ling 2
Chin Kin-Fah 3
Mahadeva Sanjiv 4
1 Department of Primary Care Medicine, University Malaya Primary Care Research Group (UMPCRG), Faculty of Medicine, University of Malaya , Kuala Lumpur , Malaysia
2 Pharmacy Department, University Malaya Medical Centre , Kuala Lumpur , Malaysia
3 Department of Surgery, Faculty of Medicine, University of Malaya , Kuala Lumpur , Malaysia
4 Department of Medicine, Faculty of Medicine, University Malaya , Kuala Lumpur , Malaysia
Lee Yeong Yeh
Electronic publication date: 2014 Jun 26
Publication date: 2014
Volume: 2
Electronic Location ID: e451
Received 2014 Apr 16; Accepted 2014 Jun 4
Copyright: © 2014 Lai et al.
Copyright year: 2014
Copyright holder: Lai et al.
License: This is an open access article distributed under the terms of the Creative Commons Attribution License, which permits unrestricted use, distribution, reproduction and adaptation in any medium and for any purpose provided that it is properly attributed. For attribution, the original author(s), title, publication source (PeerJ) and either DOI or URL of the article must be cited.
License URL: https://creativecommons.org/licenses/by/4.0/

Keywords: Intravenous proton pump inhibitor, Appropriate clinical use, Audit, Gastrointestinal bleeding

Funding: No formal funding was received for this work.

==============================
Background. Proton pump inhibitors (PPIs) are currently the most effective agents for acid-related disorders. However, studies show that 25–75% of patients receiving intravenous PPIs had no appropriate justification, indicating high rates of inappropriate prescribing.

Objective. To examine the appropriate use of intravenous PPIs in accordance with guidelines and the efficacy of a prescribing awareness intervention at an Asian teaching institution.

Setting. Prospective audit in a tertiary hospital in Malaysia.

Method. Every 4th intravenous PPI prescription received in the pharmacy was screened against hospital guidelines. Interventions for incorrect indication/dose/duration were performed. Patients’ demographic data, medical history and the use of intravenous PPI were collected. Included were all adult inpatients prescribed intravenous PPI.

Main Outcome Measure. Proportion of appropriate IV PPI prescriptions.

Results. Data for 106 patients were collected. Most patients were male [65(61.3%)], Chinese [50(47.2%)], with mean age ± SD = 60.3 ± 18.0 years. Most intravenous PPI prescriptions were initiated by junior doctors from the surgical [47(44.3%)] and medical [42(39.6%)] departments. Only 50/106(47.2%) patients had upper gastrointestinal endoscopy/surgery performed to verify the source of bleeding. Unexplained abdominal pain [81(76.4%)] was the main driver for prescribing intravenous PPIs empirically, out of which 73(68.9%) were for suspected upper gastrointestinal bleed. Overall, intravenous PPI was found to be inappropriately prescribed in 56(52.8%) patients for indication, dose or duration. Interventions on the use of intravenous PPI were most effective when performed by senior doctors (100%), followed by clinical pharmacists (50%), and inpatient pharmacists (37.5%, p = 0.027).

Conclusion. Inappropriate intravenous PPI usage is still prevalent despite the enforcement of hospital guidelines. The promotion of prescribing awareness and evidence-based prescribing through education of medical staff could result in more judicious use of intravenous PPI and dose-optimization.

Introduction

Proton pump inhibitors (PPIs) are currently the most effective agents for acid-related disorders. The high degree of acid suppression by PPIs make these drugs an ideal option in the treatment of various gastrointestinal disorders, where acid suppression promotes recovery (Brett, 2005; Metz, 2000). This is achieved by the formation of quarternary anionic structures, which then inhibits the secretion of hydrochloric acid into the stomach lumen by inhibiting the H+/K+/ATPase of gastric parietal cells (Leontiadis, Sharma & Howden, 2005). Continuous intravenous PPIs enables maintenance of an intragastric pH ≥ 6, which minimizes peptic activity and concurrently; platelet function is optimized and fibrinolysis is inhibited (Leontiadis, Sharma & Howden, 2005). These actions help stabilize clot formation over the ulcer, thus making intravenous PPIs the drug of choice for peptic ulcer haemorrhage. Studies have shown that treatment with a PPI reduces the risk of ulcer re-bleeding, thus reducing the need for surgery; but has no benefit on overall mortality (Leontiadis, Sharma & Howden, 2005).

Intravenous PPIs are indicated in the treatment of perforated gastric/duodenal ulcers, peptic ulcer disease, grade III/IV oesophagitis with bleeding and stress ulcer prophylaxis (in ventilated, critically ill patients) (Vanderhoff & Tahboub, 2002). With these new recommendations, a dramatic increase in both oral and intravenous PPI use has been observed across the globe over recent years (Gingold et al., 2004; Hoover, Schumaker & Franklin, 2009; Slattery et al., 2007). However, several studies have demonstrated that 25–75% of patients receiving PPIs, particularly intravenous preparations, had no appropriate indication (Craig et al., 2010; Hoover, Schumaker & Franklin, 2009; Slattery et al., 2007). This emerging trend is worrisome as it reflects high rates of inappropriate prescribing of PPIs in hospitals, leading to drug waste which could have otherwise been prevented (Forgacs & Loganayagam, 2008; Nasser, Nassif & Dimassi, 2010).

Several audits on the appropriateness of intravenous PPIs have been conducted in the United States (Hoover, Schumaker & Franklin, 2009; MacLaren et al., 2006), Canada (Afif et al., 2007), Europe (Craig et al., 2010; Slattery et al., 2007) and the Middle East (Alsultan et al., 2010). Some studies were retrospective (Afif et al., 2007; Chavez-Tapia et al., 2008; MacLaren et al., 2006; Naunton, Peterson & Bleasel, 2000), whilst others were prospective (Alsultan et al., 2010; Sebastian et al., 2003; Slattery et al., 2007) in study design. In addition, two qualitative studies explored the barriers and perceptions of healthcare professionals in the use of intravenous PPIs (Grime, Pollock & Blenkinsopp, 2001; Hayes et al., 2010). To date, little is known about the prescribing practice of IV PPI in Malaysia.

In one tertiary hospital in Malaysia, guidelines on the use of intravenous PPIs have been set up by the Hospital’s Drugs and Therapeutics (D&T) Committee (Fig. 1). Although pharmacists in this hospital screen all intravenous PPI prescriptions upon its receipt in the inpatient pharmacy, little is known about the usage of intravenous PPIs, nor the effectiveness of this screening process. Our hypothesis is that there may still be a proportion of intravenous PPI prescriptions that may not be prescribed according to guidelines.

Figure 1 Image of current guidelines.

Guidelines on the use of intravenous proton pump inhibitors. PPI, proton pump inhibitor; UGIE, upper gastrointestinal endoscopy; UGIB, upper gastrointestinal bleed.

Aim of the study

To assess if the usage of intravenous PPIs was in accordance with guidelines, factors associated with its use and the effectiveness of a pharmacy-led intervention.

Method

This prospective study was conducted from May to August 2010 in a tertiary hospital in Malaysia. Study patients included adult inpatients prescribed intravenous pantoprazole (Nycomed GmbH, Konstanz, Germany) since pantoprazole was the only intravenous PPI available during the period of study. Patients aged <15 years old and those prescribed only oral PPIs were excluded. Approval from the hospital’s Medical Ethics Committee was obtained prior to the commencement of this study.

Procedure

All manual prescriptions for intravenous pantoprazole received in the inpatient pharmacy were screened by pharmacists to determine if they were in accordance with hospital guidelines (Fig. 1). Interventions were performed either face-to-face by clinical pharmacists; or via the telephone by inpatient pharmacists (for areas not serviced by clinical pharmacists). During the period of study, every 4th case of a recent hospital admission prescribed intravenous pantoprazole was selected and followed-up during the duration of their stay in the hospital. Both the medication charts and clinical notes were examined to determine the rationale for prescription. Patients’ demographic data, past and current medical history and use of intravenous pantoprazole were collected using a structured data collection form. Patients were classified into two groups: those with suspected UGIB or those without (non-UGIB). All patients who had an upper gastrointestinal endoscopy (UGIE) or surgery had their reports reviewed. Stigmata of recent haemorrhage were defined as per Forrest classifications.

Definitions

The use of intravenous PPI was classified as appropriate if the diagnosis or findings (confirmed by UGIE or surgery) corresponded to the approved indications as shown in Fig. 1. If intravenous PPI was discontinued within 72 h for unapproved indications, its use was also classified as appropriate. (This decision was made by the D&T Committee to provide clinicians some flexibility.) For UGIB, intravenous PPI use was considered appropriate if there was presence of recent haemorrhage at UGIE or surgery, defined as above. Appropriate intravenous PPI dosing was defined as 80 mg bolus of pantoprazole, followed by pantoprazole infusion at 8 mg/h for 72 h (Ghassemi, Kovacs & Jensen, 2009). Suboptimal dosing regimens such as twice daily bolus intravenous pantoprazole were considered inappropriate. Use of intravenous PPI was considered inappropriate in patients with isolated variceal bleeding (MacLaren et al., 2006) and in patients too well to undergo UGIE or where UGIE was considered not necessary. For patients who were haemodynamically unstable, with haematemesis, melaena or haematochezia, the use of intravenous PPI was considered appropriate.

For non-UGIB, intravenous PPI use was considered appropriate for stress ulcer prophylaxis in critically ill patients or patients previously on oral PPI (provided they were nil by mouth). The appropriate dose would be 40 mg bolus once daily. Use of intravenous PPI in patients with abdominal pain or vomiting was considered inappropriate unless if the patient had another reason for intravenous PPI use and could not tolerate oral medications.

Each patient was followed-up until discharge or death. The following data were collected: haemodynamic status, time to initial UGIE, when UGIE was performed, operative record, duration and dose of intravenous PPI use, as well as discharge oral PPI use. Factors predicting inappropriate use were also examined: patient age, gender, ethnicity, speciality of the prescriber and prescriber status.

Statistical Analysis

Data were entered into the Statistical Package for Social Sciences (SPSS) version 18 (Chicago, Il, USA). Continuous data were expressed as mean ± SD. Categorical variables were expressed as absolute (number) and relative frequencies (percentage). Categorical data were analysed using chi-squared tests. A p-value of <0.05 was considered as statistically significant.

Results

During the period of the study, a total of 409 patients were prescribed intravenous PPI. Only 106 patients were collected according to the methodology described. Baseline demographics and clinical details are shown in Table 1. Most patients were male [n = 65(61.3%)] and Chinese [50(47.2%)], with a mean age of 60.3 ± 18.0 years [range = 15–96]. A total of 83(78.3%) patients had concurrent illness upon admission, with hypertension [n = 50(47.2%)], diabetes [n = 31(29.2%)] and heart disease [n = 24(22.6%)] being the most common problems. Sixty two (58.5%) patients were on aspirin [n = 26(25.5%)], clopidogrel [n = 12(11.3%)] and enoxaparin [n = 10(9.4%)]. The majority of intravenous PPI prescriptions were initiated by doctors from the surgical [47(44.3%)] and medical [42(39.6%)] departments; most of whom were junior doctors (medical officers without postgraduate qualifications) [n = 73(68.9%)] (Table 1). Unexplained abdominal pain [81(76.4%)] was the main presenting symptom for these patients and was the driver for prescribing intravenous PPIs empirically.

Table 1 Baseline demographics and clinical details of patients initiated on intravenous proton pump inhibitors.

Characteristics	Number (%)	Number of appropriate
intravenous PPI
prescriptions (%)	p-value	
Age (years)				
<60	40 (37.7)	26 (65.0)	0.616	
>=60	66 (62.3)	46 (69.7)		
Gender				
Male	65 (61.3)	45 (69.2)	0.717	
Female	41(38.7)	27 (65.9)		
Ethnicity				
Chinese	50 (47.2)	34 (68.0)		
Malay	30 (28.3)	21 (70.0)	0.669	
Indian	23 (21.7)	15 (65.2)		
Others (Indonesian, Nigerian, Bangaladeshi)	3 (2.7)	2 (66.7)		
Speciality of prescriber				
Surgical	47 (44.3)	34 (72.3)		
Medical	42 (39.6)	27 (64.3)		
Intensive Care	13 (12.3)	8 (61.5)	0.348	
Orthopaedics	3 (2.8)	3 (100.0)		
Obstetrics & Gynaecology	1 (0.9)	0		
Designation of prescriber				
Senior doctors (specialists)	33 (33.1)	24 (72.7)	0.476	
Junior doctors (medical officers)	73 (68.9)	48 (65.8)		
Mean duration of hospital stay ±SD (days) [range]	20.6 ±19.9 [1-109]			
Mean haemoglobin levels at admission ±SD (g/L) [range]	10.2 ±2.7 [4.4-18.2]			
<8	21 (19.8)	16 (22.2)		
8–10	32 (30.2)	21 (29.2)	0.663	
>10	53 (50.0)	35 (48.6)		
Procedure to verify source of bleeding				
Endoscopy	44 (41.5)	27 (61.4)		
Surgery	6 (5.7)	5 (83.3)	0.399	
None	56 (52.8)	40 (71.4)		

Procedure to verify source of bleeding

Only 50/73(68.5%) patients had either an UGIE [n = 44/50(88.0%)] or surgery [n = 6/50(12.0%)] performed to verify the source of bleeding (Table 1). UGIE for other patients with suspected UGIB was not performed for the following reasons: not clinically significant UGIB: n = 29(27.4%), critically ill: n = 20(18.9%), early mortality: n = 3(2.8%), recent endoscopy performed: n = 3(2.8%), and no consent obtained: n = 1(0.9%).

Among the 44 patients who had UGIE, 27(61.4%) cases were performed within 24 h and a further 17(38.6%) within 48 h. Only 1(2.1%) UGIE was performed after office hours. Most patients [n = 5(83.3%)] also had their surgery performed within 24 h from admission.

Appropriateness of intravenous PPI use, dose and duration

Overall, intravenous PPI was found to be inappropriately prescribed in 56(52.8%) patients for indication, dose or duration. However individually, 34(32.1%) patients were prescribed for an incorrect indication, 34(32.1%) were prescribed an incorrect dose and 38(35.8%) were prescribed an incorrect duration. A total of 73(68.9%) prescriptions were initiated for suspected UGIB. Within the non-UGIB group (n = 33), stress induced ulcer [n = 9(27.3%) of non-UGIB cases)], abdominal pain [n = 8(24.2%)] and post operation prophylaxis [n = 3(9.1%)] were the most frequent indications. There was no difference between the UGIB and the non-UGIB group with regards to the inappropriateness of intravenous PPI use [UGIB = 21(26.9%) versus non-UGIB = 13(46.4%), χ2 = 3.598, p = 0.058].

Intravenous PPI prescriptions among patients with an UGIB who had undergone UGIE or surgery were less appropriate than those who had not (62.2% vs. 89.3%, p = 0.012) [Fig. 2]. Similarly, with respect to the dose & duration, there was less appropriate prescribing amongst patients who had undergone UGIE or surgery compared to those who had not (42.2% vs. 85.7%, p < 0.001 and 48.9% vs. 89.3%, p < 0.001, respectively).

Figure 2 Image of appropriateness of PPI use.

*Clinically significant at p < 0.05 using the chi-square test. UGIB, upper gastrointestinal bleed; UGIE, upper gastrointestinal endoscopy.

Interventions on the use of intravenous PPIs

A total of 28 prescribing interventions were performed on the use of intravenous PPI: incorrect indication, incorrect dose and incorrect duration (Fig. 3). In one patient, pantoprazole was prescribed as an intravenous bolus dose of 40 mg three times daily. Both the inpatient pharmacist and the senior doctor intervened, but the dosage was only corrected after the senior doctor’s intervention. Interventions by senior doctors were most effective [5/5(100%)] compared to those provided by the clinical or inpatient pharmacists, respectively [8/16(50.0%) and 3/8(37.5%)], and this difference was statistically significant (χ2 = 4.91, p = 0.027).

Figure 3 Interventions performed on the use of intravenous proton pump inhibitor.

There were other issues that required intervention: intravenous PPI was prescribed in 34/106(32.1%) patients where its use was not justified, but interventions were only performed in 20/34(58.8%) patients. Three patients were started on the incorrect dose of intravenous pantoprazole: (i) 40 mg bolus loading dose (ii) 40 mg bolus dose administered three times daily and (iii) an incorrect dilution of 80 mg in 40 mL normal saline at 8 mL/hour for the high infusion dose. Prescribers also failed to convert 55(51.9%) patients from intravenous to oral PPI once the patient was clinically well enough to start oral intake.

Discussion

This study was conducted in a tertiary hospital over a 14-week period to assess the usage of intravenous PPI and its adherence to hospital guidelines. It was found that intravenous PPI was inappropriately prescribed in 52.8% patients, affirming our initial hypothesis that a number of doctors were prescribing intravenous PPI defensively in situations where unexplained abdominal pain was the main driver for inappropriate therapeutics. This could be due to the fear of liability arising from allegations of under-treatment, creating an error of commission rather than an error of omission. However, there is a price to be paid for defensive prescribing. The cause of the abdominal pain may not be as thoroughly investigated and unnecessary use of intravenous PPIs escalates total cost.

The decision to prescribe is influenced by many factors, such as the doctor’s perceptions of the patient’s social background, beliefs, attitudes and expectations, as well as the uncertainty of the diagnosis (Greenhalgh & Gill, 1997). In addition, the lack of knowledge of the specifics on how to manage UGIBs and limited belief in the value of guidelines especially in areas where evidence is lacking (e.g., in Intensive Care Units) may influence a clinician’s decision to prescribe inappropriately. Variability of knowledge and skills of junior and senior healthcare professionals together with a limited concern regarding cost or side effect implications could potentially be the other barriers (Hayes et al., 2010).

Overall, the inappropriate use of intravenous PPI in the present study (52.8%) were lower than findings from other studies which ranged from 57 to 78% (Afif et al., 2007; Craig et al., 2010; Hoover, Schumaker & Franklin, 2009). One possible reason could be because there were no existing guidelines in the other hospitals whereas an existing guideline plus a pharmacy-led intervention was already in place in our present study. Inappropriate use was most common in non UGIBs (Craig et al., 2010), but we did not find this difference in our present study (which may be due to the small sample size). The leniency of our definition of “appropriateness”, whereby a leeway of prescribing intravenous PPI empirically for 3 days before discontinuation, may have influenced our data. Some studies have also shown that inappropriate intravenous PPI prescribing was strongly associated with surgical admissions (Craig et al., 2010; Hoover, Schumaker & Franklin, 2009) and prescriptions initiated by junior doctors (Craig et al., 2010). However, it is a practice in our hospital that most junior doctors are prescribing based on advice given by the senior doctor, indicating that a gap of knowledge exists at both levels.

This audit found higher rates of inappropriate intravenous PPI use among patients with an UGIB who had undergone UGIE than those who did not (62.2% versus 89.3%, p = 0.012). These findings are contrary to other studies which showed that there was higher association between appropriate uses of IV PPI with respect to UGIE (Alsultan et al., 2010). The high rate of inappropriate use in the present study was due to the doctor’s failure to stop intravenous PPI therapy once findings were confirmed to be Forest 3 gastric/duodenal ulcers (90.0%), variceal bleeds (62.5%) and negative UGIE outcomes (54.5%). Intravenous PPI was also prescribed at an incorrect dose and duration more often in patients who had undergone UGIE or surgery compared to those who had not (42.2% vs. 85.7%, p < 0.001). These findings are higher than expected when compared to other studies (Afif et al., 2007). As most of these cases had UGIB, the complete intravenous PPI regimen for UGIB (bolus loading dose of 80 mg, followed by a high dose infusion at 8 mg/h for 72 h) was not prescribed. The number of patients who received the loading dose followed by intravenous infusion was very low. The lack of the intravenous 80 mg bolus dose in these patients could have delayed acid suppression and might constitute a possible dosing error (Brunner et al., 1997). In patients who did not undergo UGIE, most patients were prescribed 40 mg bolus twice daily—the most commonly prescribed intravenous PPI dose. These were appropriate doses as these patients did not have suspected UGIB.

Early UGIE allows for safe and prompt discharge of low risk patients, improves outcomes for high risk patients and reduces resource use (Hayes et al., 2010). This audit revealed that UGIE/surgery was only performed in 50 (47.2%) cases with suspected UGIB. Whilst some of the reasons for withholding UGIE appeared valid (i.e., too critically ill or early mortality), a number of patients had no evidence of clinically significant UGIB, usually a suspected benign condition like Mallory-Weiss tears. Whilst the clinicians managing these patients were confident enough to withhold an UGIE, the continuation of intravenous PPI in these cases suggested their lack of experience in managing these cases.

The screening process by the pharmacy department for intravenous PPI prescription was inadequate in this study. A number of prescriptions for intravenous PPI arrived after office hours and bypassed the usual screening process by pharmacists. The pharmacy technician on duty supplied one day’s treatment of intravenous PPI. The prescription should then have been sent to the inpatient pharmacy the following day to be screened in the usual manner. However, these prescriptions are sometimes “lost” in a paper trail and pharmacists may fail to screen these prescriptions. The knowledge and application of guidelines may not be optimally monitored in a hospital for different reasons: the healthcare professionals’ lack of awareness of a monitoring process, a lack of formalized monitoring process, or the unwillingness of some pharmacists to challenge a doctor’s prescribing behavior (Hayes et al., 2010). Interventions on the use of intravenous PPI were most effective when performed by senior doctors (100%), followed by clinical pharmacists (50.0%), and least effective when performed by inpatient pharmacists (42.9%). This finding was as expected, as junior doctors were more likely to follow the advice of their seniors. In our hospital, only 7/33 (21.2%) wards have clinical pharmacists. Clinical pharmacists are more effective in their interventions as they have face-to-face contact and a working relationship with doctors on the ward. Inpatient pharmacists were intervening over the phone. This type of intervention is impersonal and tends to be ineffective. Possible solutions to this problem include an order template, and to have more clinical pharmacists to cover wards.

This study has several limitations. Although systematically selected, our cases for study may have not been entirely representative of all patients administered intravenous PPI in this institution. Only one out of every fourth prescription was selected due to time constraints. Data collection over a longer period of time (either 6 or 12 months) would have minimised this limitation. Secondly, definitions of appropriateness used in this study may not have been entirely consistent with other publications on this topic. Our definitions were derived largely from decisions made by the D&T committee.

Conclusion

Inappropriate intravenous PPI usage is still prevalent despite the enforcement of hospital guidelines. The promotion of prescribing awareness and evidence-based prescribing through education of medical staff could result in more judicious use of intravenous PPI, not only in terms of approved hospital indications but also in dose-optimization according to indication. Ward-based clinical pharmacists have a role, but have less of an impact on changing prescribing errors when compared to senior doctor intervention.

Supplemental Information

Supplemental Information 1 Raw dataset for IV PPI study

Click here for additional data file.

We would like to thank Siew Chen Lim, Huey Shean Lee, Han Chen Low and the nursing staff (of the surgical and medical wards) for their assistance in data collection.

Additional Information and Declarations

Competing Interests

Author Contributions

Human Ethics

Dr. Sanjiv Mahadeva has received professional speaker’s fees from Astra Zeneca. Dr. Chin-Fah Kin has received speaker’s fees from Astra Zeneca Malaysia.

Pauline Siew Mei Lai conceived and designed the experiments, analyzed the data, wrote the paper, prepared figures and/or tables, reviewed drafts of the paper.

Yin Yen Wong, Yong Chia Low and Hui Ling Lau performed the experiments.

Kin-Fah Chin contributed reagents/materials/analysis tools.

Sanjiv Mahadeva analyzed the data, contributed reagents/materials/analysis tools, wrote the paper, reviewed drafts of the paper.

The following information was supplied relating to ethical approvals (i.e., approving body and any reference numbers):

The University Malaya Medical Center’s Medical Ethics Committee: IRB ref no. 793.15.

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
