# Peer review of "Unexplained abdominal pain as a driver for inappropriate therapeutics: an audit on the use of intravenous proton pump inhibitors"

_PeerJ, doi:10.7717/peerj.451_

## Round 0.1 · original submission · Minor Revisions

A well-written paper addressing an important issue on proton-pump inhibitor's prescription. Comments from reviewers are generally favourable although there are some concerns which will need review. I look forward to a revised version of the paper.

Reviewer 1 ·

Basic reporting

Inadequate referencing.
In the introduction section, the authors state that "25-75% of patients do no have an appropriate indication." This should be referenced.

They also state that 2 qualitative studies explained the barriers and perceptions.." this should also be referenced.

Also in the discussion section, the authors state that "overall, the inappropriate use of intravenous PPI in the present study (52.8%) were
239 lower than findings from other studies which ranged from 57-78%". What is the reference for this?

There is a very recent study published in the Singapore Medical Journal by Chris Chia et el., on the same topic of inappropriate PPI use. This should be included in the reference list.

Discussion section is rather repetitive.

Experimental design

Who performed the data collection?
In cases where the reason was not clear from the case notes about PPI use, was the primary physician contacted? This should be mentioned in the methods section.

When the authors mention that PPI use was prescribed by a junior doctor, could the junior doctor have prescribed the PPI under the instruction of the senior doctor? Is the method of prescription in the hospital electronic or manual? Did the pharmacist retrieve the prescribing doctor's information based on an electronic prescription or manual prescription? Most times in real life situations, the junior doctor writes the prescriptions or does an electronic prescription based on advice from a senior doctor. Could this have been the practice in the hospital? If so, then we can't conclude that inappropriate use of PPI was due to junior doctors prescribing PPIs.

Validity of the findings

Overall the paper is fair but lacking in novelty.
There have been several reports about inappropriate PPI use in various countries. The authors need to fine tune their reference lists.

Reviewer 2 ·

Basic reporting

In general, it will be good to quote references as in standard format or bracketed at end of each statement.

line:114 - two classified groups mentioned (with UGIB & non-UGIB) TOTAL n=106
Fig 2: suspected UGIB (n=73)
With regards to the above; is this another group under "suspected UGIB"?
WHAT happened to the other 33 samples? (106-73=33)

Table 1: Mean Hb at admission 10.2+/- 2.7 (4.4-18.2)
Suggest that Hb levels be classified to various ranges to be more meaningful.
Eg. <8 / 8-10 / >10 or have different range for male & female.
This will also reduce the effect of extreme values at both ends eg 4.4 & 18.2 attributing to the mean & SD.

Experimental design

Sample size is too small. Hence the subgroup has denominators which is single digit ; examples ; line 205 (5/5) & line 206 (3/8) . Yates correction may have been used here ; however; it will be better if a total sampling were performed & over a longer duration since this is more of an audit, cost may not be a hindrance.
In year 2003 ; the incidence rate of UGIB in Malaysia was 72/100,000 population (peaked at 4th-6th decade) with mortality rate of 10.2% (increasing substantially with age and no difference between gender)
Using the above data, sample size required need to be larger.

Validity of the findings

As mentioned, findings are relevant and will be more representative if data collection were over a longer duration & a total sampling will reflect a tertiary hospital better.

Note that the group which did NOT have procedure performed was having less "inappropriate use/dose/duration" ; as this result may not be reflective of the true nature because the some of the diagnoses may not be accurate. (no UGIE done)
However the number of cases with doubtful diagnosis can be small.

No new information from this study, unless the study can provide some related causal factors & include innovative solutions to the problems identified.

Additional comments

no comments

---

## Round 0.2 · accepted · Accept

The revised version has been reviewed. Concerns raised by reviewers have been adequately addressed by authors. Although the study may not be novel and there is a limitation in sample size but the underlying message is important. It is time to reflect on the prescribing practice of proton-pump inhibitors in the local context.